# Comparison between Different Mechanization Systems: Economic Sustainability of Harvesting Poplar Plantations in Italy

**Valerio Di Stefano** [1,2], **Giorgia Di Domenico** [1,2,*], **Michele Menta** [1], **Elisa Pontuale** [2], **Leonardo Bianchini** [1] and **Andrea Colantoni** [1]

1   Department of Agriculture and Forest Sciences (DAFNE), University of Tuscia, Via S. Camillo de Lellis snc, 01100 Viterbo, Italy; valerio.distefano@crea.gov.it (V.D.S.); l.bianchini@unitus.it (L.B.); colantoni@unitus.it (A.C.)
2   Council for Agricultural Research and Economics, Research Centre for Forestry and Wood, Via Valle della Quistione, 27, 00166 Rome, Italy
*   Correspondence: giorgia.didomenico@crea.gov.it; Tel.: +39-06-47836241

**Abstract:** After a period of significant development, poplar cultivation in Italy has been in rapid decline since the 1980s. Because of its marked ductility, poplar is valuable for both wood furniture and energy production. Production could be increased through mechanization, because innovative machinery and equipment can reduce the exposure of forest workers to common risk factors, ensure greater and better productivity, increase the efficiency of operations, and reduce costs. There are various systems for the mechanization of poplar production (from traditional to advanced and pushed mechanization). We describe the range of possibilities (in terms of both the techniques adopted and the machines used) for planting, harvesting, and chopping poplar. Based on our analysis of operating costs, we conclude that mechanized poplar production could reduce the average cost per ton of wood chips (EUR/t) by 23% and the average gross cost per hectare of wood chips produced (EUR/ha) by 37%.

**Keywords:** poplar cultivation; mechanization; harvesting systems; economic sustainability

## 1. Introduction

In Italy, the management of forest stands of natural origin is increasingly oriented towards the promotion of functions of a non-speculative nature and of public environmental interest. In this context, the sustainable production of wood from poplar trees plays a significant role in storing carbon dioxide from the atmosphere, sequestering it in long-term works (such as furniture, building materials, etc.). This practice contributes to the pursuit of collective well-being, promoting climate change mitigation and environmental and landscape improvement in rural areas.

While, on the one hand, the environmental functions can be entrusted more with the natural forest heritage, on the other hand, the economic productive function is increasingly delegated to arboriculture plants. These plants provide a positive response to the needs of accelerated production (with increases not reachable in the forest), the possibility of better planning, and the search for consistency and homogeneity of the dimensions and technological characteristics of the assortments obtainable [1], while carrying out in part the environmental functions delegated to the natural forest patrimony. In this regard, the common agricultural policy (CAP) provides economic incentives for the cultivation of more environmentally sustainable poplar hybrids (MSA), at the suggestion of the "From Farm to Fork" strategy [2], with the aim of contributing to carbon neutrality by 2050 [3].

The poplar cultivation sector in Italy is undergoing an important evolution, both in technical and economic terms: from 2018 to today, the price per ton of poplar wood has increased by 50% from EUR 45 to 50 per ton to the current EUR 100–110 per ton.

The reasons for this evolution are many, and the most important is certainly the ductility of poplar wood: soft, whitish and light, it lends itself well to being worked, glued, and painted. It can be used in various economic and production sectors such as carpentry (packaging, scaffolding elements, wood wool, plywood, chipboard, matches, and occasionally furniture) and in the paper industry [4]. For outdoor use, poplar wood is usually heat-treated. Moreover, poplar is a renewable energy source; it is among the most efficient tree species, as—more than others—it optimizes the production yield of biomass that is used to produce thermal and electrical energy.

However, a careful analysis of the state of poplar cultivation is difficult today due to the progressive reduction in the information base, not only on the extent of cultivated areas, production, and the number of operators, but also economic variables directly linked to production (timber prices, costs of cultivation operations, concession fees, etc.). It is true, however, that the growing increase in labor costs in recent years has not been fully offset by an increase in the market value of the timber product, and for the entrepreneur to maintain their business, they are necessarily forced to keep costs as low as possible. The costs derived from crop production contribute significantly to the final cost of the biomass [5,6] and there are many aspects of the agronomic process that could be optimized through applying science and innovation. Therefore, it is necessary to identify these aspects correctly as a first step to finding alternative solutions [7,8]. To increase the economic sustainability of poplar weights, productivity, and therefore the competitiveness of the material on the market, it is certainly necessary to strengthen and improve the level of mechanization by encouraging the use of more innovative machines [9].

The benefits of mechanization development from the 1980s to the present are evident [10]: an increase in the level of mechanization offers advantages to wood workers not only in reducing costs and increasing the productivity of the yard [10], but also in ensuring greater worker safety [11–13]. In the present work, three possible levels of mechanization to be used in the different phases of the forest worksite (felling/staging, harvesting, shredding, and loading) are distinguished and analyzed: (i) traditional, (ii) advanced, and (iii) pushed.

While at the traditional level, the use of a chainsaw is mainly expected, at the advanced one, the abatement occurs with a head of abatement (e.g., pliers with clamping arms and cutting device at the base, mounted on a hydraulic articulated arm of a machine with sufficient power, normally >80 kw), thus leaving the staging phase to the chainsaw. The integration of mechanical means takes place in heavy mechanization in which a series of combined saws called combined harvesters (both for felling and for setting up) are used.

For the harvesting phase, more traditional means, such as the use of a tractor with the addition of a winch (traditional and advanced mechanization) [14], have been replaced by more advanced means, such as by a forwarder or an articulated trolley and a skidder or an articulated tractor equipped with pliers and/or winch.

The aim of this work is to analyze the main mechanization techniques present in the field of poplar cultivation, deepening the multitude of machines used in forestry processing and devoting ample space to the analysis of the operating costs of equipment in a comparative perspective with respect to the most effective and sustainable level of mechanization.

The following techniques were used for this review of the scientific literature: the main search engine used was the Scopus and Science Direct database. Keywords such as forestry mechanization, poplar cultivation, short rotation coppice, harvesting systems, and operating costs were used for the research. There was also bibliographical research at the library of the Council for Research in Agriculture and Economics (CREA) in Rome and Casale Monferrato (AL). The selection criteria for the articles for this review focus on the year of publication, preferring the most recent scientific publications except for the issue of operating costs, as there is no recent literature on this subject, and on the consistency with the main species under analysis, namely poplar. Articles with a similar type of cultivation of forest species and category of use of final biomass were also considered.

## 2. Poplar Cultivation in Italy

The poplar cultivation in Italy is a sector of excellence to produce wood for industrial and energy use [4]. Wood arboriculture occupies about 100,000 ha [15] and the sector is dominated by poplar cultivation that, compared to the total forest area, provides half of the annual harvests of wood [16]. Covering less than 1% of the national forest area, 46,125 ha in total, it supplies about 1 million m$^3$ of round wood per year, equal to 45% of the domestic round wood [17]. The production area is located for 94% in the Po Valley with prevalence in Lombardy and Piedmont (70%) for a total of over 32,000 ha of poplar. More than 50% of the area falls in floodplain areas, just under 30% in Special Protection Areas (SPAs), while the remaining 20% is in Sites of Community Interest (SCI) and in park areas [4].

It should be noted that since the period 1950–1970, there has been a phase of great development of Italian poplar cultivation, the extension of which has reached over 170,000 ha of plantations. However, since the 1980s, poplar plantations have experienced a period of marked and constant decline until they reached less than half of the initial values, or about 83,000 ha [18].

According to the last National Forest Inventory [19], the Italian area of artificial poplar in 2015 is equal to 51,592 hectares with a total surface loss of 14,678 ha compared to 2005. According to [4], other estimates made on a sample basis in 2018 with reference to poplar weights of a surface equal to or greater than 5000 m$^2$, excluding polycyclic poplar plantations, reported that the total area of poplar trees in Italy is equal to 46,125 ha. Considering the above, the estimates underline the already known phase of decline with a loss of about 40,000 ha compared to the beginning of the year 2000. The following histogram (Figure 1) shows the estimated regional area (ha) for poplar cultivation in Italy in 2015 and 2018.

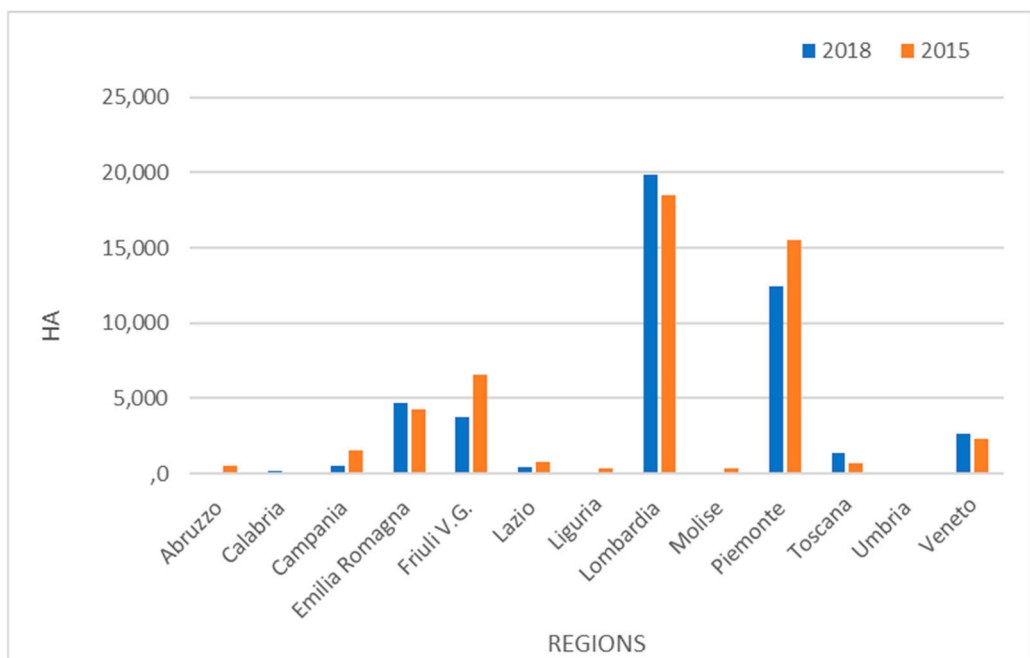

**Figure 1.** Regional estimated area (ha) for poplar cultivation in Italy in 2015 [19]–2018 [4]. The histogram shows only those regions with poplar cultivation with an extension of more than 50 ha in the years considered.

## 3. Mechanization in Plant Systems

The short rotation coppice (SRC) is a cultivation system that has been adopted for about thirty years in Italy for fast-growing species [20] such as poplar. It is used, generally, to produce biomass (wood chips for energy uses or as industrial material). This model provides a high level of density (5000–10,000 plants per hectare) and yield shifts every

2–3 years [21–25]. By applying longer shifts (5–7 years), the model is also referred to as medium rotation coppice (MRC) [20,26]. In the latter case, the density is about 1600/ha plants and the aim is the production of biofuel [27].

Finally, the traditional planting system provides for 8–12-year shifts and cultivation densities of 280–300 plants per hectare, and the final product is good-quality wood for industry and to produce products such as paper, sawn timber, and panels [28].

The economic viability of the plant depends mainly on minimizing the cost of production of the final product, especially in the case of SRC and MRC, taking into account the low commercial value of wood chips and the competition that this has on the market (there are much cheaper materials such as sawmill waste). This reduction in the cost of production can only be achieved through the complete and efficient mechanization of field operations; among these, the most expensive are certainly harvesting and planting [14,29,30].

Starting from the plant phase, there are different models of machines that differ in the type of transplant organs, the type of feed, or the type of plant material used. In the case of the SRC, and therefore of a cultivation system characterized by a sixth of the "dense" plant, the best propagation material is the cutting, while for a sixth of the "sparse" plant, generally used in the MRC or in a traditional plant, the rod is used (pegs of about 2 m) [31].

For the planting of the cuttings, it is possible to mechanize the whole process using machines capable of operating a mechanized burial. The machines to be used are: Rotor planter and Spapperi discs transplanter (Figures 2 and 3).

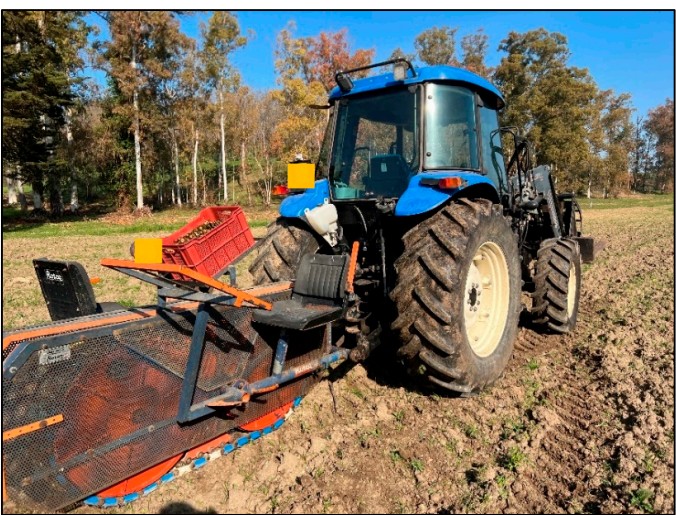

**Figure 2.** *New Holland* TD5050 crawler *Rotor* planter in experimental farm "Ovile" in Rome.

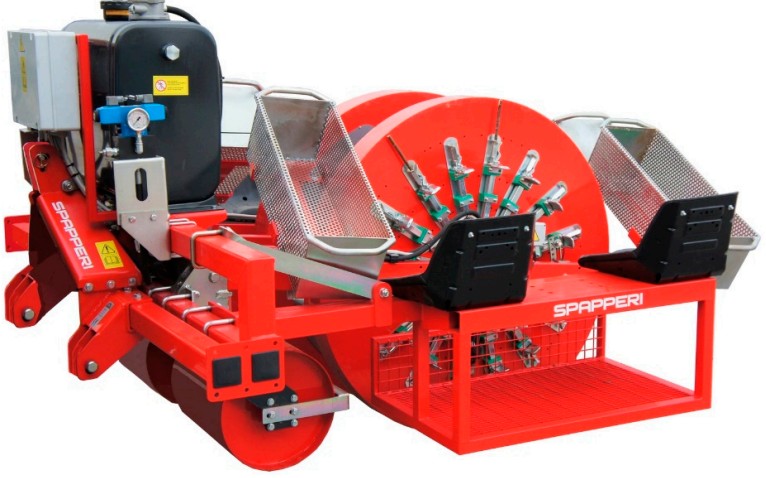

**Figure 3.** *Spapperi* hydraulic cuttings transplanter.

The first machine consists of a metal track with meshes fixed inside another cylinder, through which a piston operates. Thanks to the traction of the engine, this piston slides down, allowing for the insertion of the cutting into the cylinder. This system allows us to progressively insert the cutting vertically into the ground, whereby the planting takes place without the opening of a groove and without the air remaining around it. The *Rotor* is brought with a three-point linkage of tractors of at least 40 kW of power (55 HP) and for its operation, it requires three operators. The working speed is about 0.8–0.9 km/h with a labor requirement of 11.5 h/ha. The working capacity is 0.25 ha/h, with a productivity of about 1250/1500 cuttings/ha/employee.

The second is equipped with clamp injectors that carry out the transplantation, arranged radially and equidistant from each other on a large metal wheel that moves due to the drag caused by the movement of the tractor [32]. With the rotation of the wheel, when an injector element is found in a vertical position and perpendicular to the ground, a special proximity switch, through a solenoid valve, controls the extension of a hydraulic jack that, in turn, pushing the injector element down, inserts the cutting into the soil. The machine must be coupled to a tractor of at least 90 kW and its operation requires two operators. The working speed is about 1.4–1.6 km/h with a labor requirement of 3.5 h/ha. The working capacity is 0.87 ha/h, with a productivity of about 1750–1800 cuttings/ha/employee.

The rotor transplanter inflicts the cuttings vertically, while the disc transplant performs less accurate work, but is also able to mulch along the row with a plastic film with the dedicated module.

In the case of bolts, the cutting of the ground is carried out by means of a drill; the planting is performed manually by two operators; and the recoating is carried out by means of inclined ridgers (spreader) and compacting rollers. Now, there are only prototypes that are not yet on the market, such as the "R-Innova 500P", a machine developed on behalf of R-Innova by the agro-mechanical services company situated in Stagnati di Ostiano (Italy). These vehicles are also equipped with a platform for the transport of the material and an acoustic or light signal, electrically controlled by a device, indicating the transplantation distance [33].

In light of the above, the use of a semi-automatic cuttings transplanter, compared to the traditional manual or mechanization techniques of the first level, guarantees a significantly reduced working time and a limited use of labor [32].

## 4. Mechanization in Harvesting Systems

Harvesting systems are an essential component of woody biomass supply chains [34] and represent one of the most expensive processes along the entire production chain [35,36]. Poplar harvesting systems can be distinguished based on different parameters such as the density of the plant, the size of the trees, the space for handling the machines, the cultivation system, as well as the production purpose and the level of mechanization.

Generally, in the European Union's poplar yard, mechanization follows the custom-made harvesting model, in which the trees are taken in commercial lengths per log (cut-to-length—CTL). This is a fast and productive cutting method [37,38] that involves the use of a harvester that fells and prepares the logs in lengths generally of 2 m or 4 m, and then these are transported to the forest landing, through a forwarder. The use of CTL technology offers several advantages, in particular, an accurate evaluation and measurement of individual trees directly on the felling field, which is reflected in a better quality of the timber in terms of reducing the contamination of the logs, quality of uprooting, and recovered value [27,39,40].

At the same time, however, the CTL is relatively expensive and designed for processing single trees, which does not fit well with the limits imposed by plantations of small trees [40]. An alternative method is the harvesting system of the whole tree (whole-tree harvesting—WTH) [41], which consists of: (i) felling of whole trees, (ii) dragged by a skidder grapple on the landing, and (iii) transformed into trunks [27,39,42].

As can be seen, the WTH method requires three steps instead of the two required by the CTL method; however, the workflow is sustained at higher speeds, and the processing of trees, once on the ground, minimizes the influence of trunk size on extraction efficiency [27,43]. The WTH method, already particularly appreciated in the USA, is much more effective in terms of productivity in the felling and extraction operation, especially in the presence of constraints due to the reduced size of the trees, where the previous CTL method is unfavorable.

From an economic point of view, the WTH system provides a 10% higher productivity than the CTL system and a cost reduction of around 15 EUR/t [39]. However, the two methods shown are poorly specialized harvesting systems for medium-shift-coppice crops [44–46].

Two other harvesting systems exist and are widely diffused in the poplar trees of North America: system-to-peripheral accumulations and system-to-centralized accumulation.

The first involves felling and setting up by a dedicated machine characterized by a shear head; then the material is transported with a wheel loader to the landing where the woodchipper–root–debarker unit (CSS) is located. The centralized storage system, instead, provides the transportation of the felled plants to the storage area through a disc head mounted on a dedicated forestry tractor. Subsequently, they are transported with a forwarder to the roadside and divided according to the type of retractable assortment, sawn timber or discarded selvedges and/or small plants, whereby the latter are chipped through the CSS. Both work systems can produce more than 400 tons of biomass per day, with a total cost of EUR 12–24 per ton of biomass [46].

For the harvesting of young plantations, the only economically sustainable solution is multi-shaft handling (MTH), where more than one tree is collected per work cycle, compensating for small volumes of material [12,41]. In this case, the felling is made with a mini-crawler blast chiller characterized by a head equipped with a disc saw and accumulation arm, able to break down and accumulate two plants per work cycle, and concentrating six rows of plants on a single swath. This system is efficient and economical, having the advantage of using inexpensive and extremely mobile vehicles [46].

## 5. Mechanization in Chopping Systems

In recent years, the increasing use of alternative energy sources as opposed to fossil fuels and the consequent creation of power plants or small plants for the production of thermal energy has diversified the use of wood waste products. This is also in light of the main European policies and legislative acts enacted by the European Union itself, the European Green Deal and REPowerEU, which are also regarded as incentives for the use of renewable energy sources for greater environmental and energy sustainability [22,31]. The biomass produced by the chipping of short-rotation woody crops (SRWCs) is considered one of the most interesting options to generate renewable energy [47]. The production of wood chips also ensures the recovery of an average of 20–30% of biomass, which would otherwise be a residue of utilization [46]. With the chipper, it is possible to reduce the size of the original woody biomass up to a size of $2 \times 2$ cm; the larger the original product, the harder the machine will work, and it will take longer to chip. The poplar, due to the small size of its trunk and branches, is one of the forest species most suitable for chipping [48].

The lack of knowledge on harvesting [49] and the uncertainties regarding the expected costs and profits [50,51] are the main reasons why farmers hesitate to establish SRWCs [52].

In general, two different chopping systems are used for SRWCs: cut-and-store and cut-and-chip [36], as shown below (Figure 4).

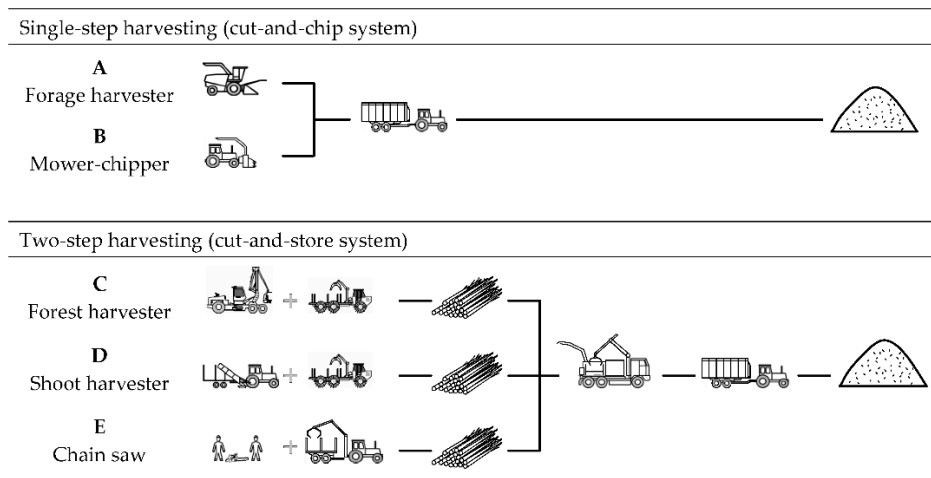

**Figure 4.** Cut-and-chip system (**A**,**B**) and cut-and-store system (**C**–**E**) [53].

The first system is realized in two phases that take place physically and are temporarily separated [26]: (i) harvesting and (ii) hauling and chipping the cut stems [54]. The harvesting systems, discussed in the previous paragraph, can be performed in a manual [55] or mechanical way. The manual harvesting is not interesting unless there are conditions for which the mechanized system cannot be used (e.g., reduced size of the plant) [36]. Mechanized harvesting operations are performed by using a specialized harvesting head attached to an agricultural vehicle that allows for the harvesting and transport in the storage place such as a dedicated site to intervene later [56] or within the plantation itself (swath) [26].

According to a study conducted in Belgium on the harvesting of a short-rotation poplar plantation [36], it was found that in the cut-and-store system, the manual harvesting is performed much more slowly than the mechanized one (0.01 vs. 0.37 ha h$^{-1}$), resulting in lower productivity (0.15 vs. 8.84 t h$^{-1}$) and higher costs (8688 vs. 779 EUR ha$^{-1}$).

In the cut-and-chip system, instead, felling and chipping occur simultaneously by using mower-feeders that directly unload the wood chips in special trailers, thus facilitating transport. According to [36], the cut-and-chip harvesting system is intermediate in terms of performance and productivity compared to the cut-and-store system: the hourly cost is lower for the manual cut-and-store system than that which is mechanized (440 vs. 674 EUR h$^{-1}$), and this does not compensate for the higher cost per hectare and ton (10.142 vs. 2232 EUR ha$^{-1}$ and EUR 426 vs. 94 EUR t$^{-1}$). For clarity, it is specified that the values considered for manual harvesting are 0.01 hah$^{-1}$ and 0.15 th$^{-1}$, while for mechanized harvesting, they are 0.37 hah$^{-1}$ and 8.84 th$^{-1}$.

Upon evaluating the relevant Italian literature [30,57–60], some harvest parameters calculated on short-rotation poplar plantations have been synthesized and are shown in Table 1 under both the mechanized cut-and-store system and cut-and-chip harvesting system. The mechanized cut-and-store chipping system (using a Jenz Hem 561) shows an average productivity of 13.90 th$^{-1}$ and an average cost per hectare of 1594.5 (EUR ha$^{-1}$); the cut-and-chip system instead shows an average productivity of 15.98 (th$^{-1}$) and an average cost per hectare of 475.08 (EUR ha$^{-1}$). For the latter case, the self-propelled forage harvesters (SPFHs), commonly called self-propelled mulchers, that is, vehicles equipped with a very specific cutting head, are Valmet 921 + 840 and Jenz HEM 561, GBE-1 e GBE-2, Claas HS-2, and HTM 1500 e Biopoplar, among which the most important versions were developed by Italian companies [26,52,57].

**Table 1.** Literature survey of harvesting operations of poplar cultivations in short rotation. Relevant parameters of the mechanized cut-and-store and cut-and-chip harvesting systems are presented. Data were collected from various studies reported in the literature [36].

| System | Plant Density (ha$^{-1}$) | Above-Ground Age (y) | Stocking Biomass (tha$^{-1}$) | Performance (hah$^{-1}$) | Productivity (th$^{-1}$) | Cost per Hour (EUR h$^{-1}$) | Cost per Hectare (EUR ha$^{-1}$) | Cost per Ton (EUR t$^{-1}$) | Machine | Reference |
|---|---|---|---|---|---|---|---|---|---|---|
| Cut and Store—Hauling | | | 51.33 | 0.23 | 5.64 | 70 | 304 | 12 | Valmet 840 | [61] |
| Cut and Store—Chipping | | | 25.49 | 0.22 | 13.63 | 340 | 1531 | 25 | JENZ HEM 561 | [60] |
| | | | 31.15 | 0.20 | 14.09 | 340 | 1681 | 24 | JENZ HEM 561 | |
| | | | 25.85 | 0.20 | 12.37 | 340 | 1704 | 27 | JENZ HEM 561 | |
| | | | 30.01 | 0.23 | 15.54 | 340 | 1462 | 22 | JENZ HEM 561 | |
| Cut and Chip | 1670 | 5 | 91.00 | 6.03 | 7.87 | 110 | 664 | 14 | Valmet 921 + 840 and Jenz HEM 561 | [60] |
| | 1670 | 5 | 81.50 | 5.51 | 9.22 | 110 | 606 | 12 | Valmet 921 + 840 and Jenz HEM 561 | |
| | 1670 | 5 | 91.00 | 6.37 | 8.26 | 110 | 700 | 13 | Valmet 921 + 840 and Jenz HEM 561 | |
| | 1670 | 5 | 81.50 | 5.97 | 8.83 | 110 | 656 | 12 | Valmet 921 + 840 and Jenz HEM 561 | |
| | 14,100 | 2 | 21.90 | 1.30 | 16.86 | 322 | 418 | 19 | GBE-2 | [57] |
| | 7100 | 2 | 16.70 | 0.60 | 27.86 | 552 | 330 | 20 | GBE-2 | |
| | 10,000 | 1 | 8.58 | 0.90 | 7.68 | 203 | 227 | 26 | Claas HS-2 | [59] |
| | 6000 | 2 | 19.60 | 0.82 | 16.08 | 234 | 285 | 15 | GBE-1 | |
| | 9000 | 3 | 24.30 | 0.46 | 11.14 | 270 | 589 | 24 | HTM 1500 | |
| | 6000 | 3 | 30.20 | 0.66 | 19.96 | 210 | 318 | 11 | Biopoplar | |
| | 6000 | 3 | 24.00 | 0.79 | 18.86 | 210 | 267 | 11 | Biopoplar | |
| | 6000 | 2 | 27.70 | 0.53 | 14.55 | 210 | 400 | 14 | Biopoplar | |
| | 6000 | 2 | 32.20 | 0.28 | 8.88 | 210 | 716 | 24 | Biopoplar | |
| | 5550 | 2 | 20.90 | 0.93 | 22.57 | | | | GBE-2 | [57] |
| | 5550 | 2 | 33.70 | 1.20 | 27.97 | | | | GBE-2 | |
| | 9520 | 2 | 31.00 | 1.06 | 29.14 | | | | GBE-2 | |

The following cost analysis refers only to a chipper on forwarders and therefore to a cut-and-chip harvesting system.

## 6. Economic Analysis of the Construction Forest Site

Below is a reworking of costs (EUR/h) made by the authors to estimate the operating costs of the machines at present. Unfortunately, it has not been possible to calculate the productivity of the machines (ton/ha), but for completeness, these data were taken from references, as presented in the previous paragraphs.

The economic evaluation is aimed at quantifying the hourly costs of the equipment and machines considered in a hypothetical traditional cultivation system and in an SRC system.

The mechanization levels examined are as follows:

(1) Traditional mechanization;
(2) Advanced/pushed mechanization.

In the first case, the following equipment/machines are considered: GS 630 standard 2T chainsaw (felling and equipment), in the case of traditional mechanization, tractor "Fendt 600 Vario" with drum winch (wood), and an 88 kW self-propelled loader (load).

In the second case, i.e., for the SRC and therefore with extensive mechanization, the following equipment/machines are considered: the harvester "Timberjak 1270 C Advance" (felling and preparation of the material), the forwarder "Komatsu 845" (logging), the "Erjo" chipper mounted on the forwarder (shredding) and, finally, for stacking, an 88 kW self-propelled loader.

The following table (Table 2) lists all the technical and economic elements used to calculate the hourly cost of using the machines. The technical values (nominal power, useful life, annual use) and the economic values (e.g., new value) were extrapolated from the data sheets of each machine operator. For the other estimated elements, the calculation equation used in the last column of the following table was inserted. In regard to average fuel costs, 1.89 EUR/L was considered for petrol and 1.45 EUR/L for agricultural fuel. Please note that the table below also shows the costs of using a Feller buncher with the wheel loader "Komatsu WA80M-8" for advanced mechanization, but that this has not been considered for the subsequent estimates of effectiveness and productivity of the system.

The elaboration of the operating cost of the machines (values in EUR/hour, Table 3) was carried out by proposing some technical coefficients deemed more suitable [62] and by applying an analytical methodology that distinguishes fixed and variable costs [62,63] as well as by applying mathematical formulas found in the main calculation methodologies proposed by different authors [64–67]. Relative to the cost of the labor, it has been considered that a single gross tariff correlates to 15 EUR/hour, inclusive of all the insurance and provident and assistance burdens previewed by the Italian legislation [18].

For greater clarity in the assessment of the economic sustainability of the machines, the productivity values given in reference [13] were considered. Based on the average gross labor productivity $(th^{-1})$ for individual operations, the authors have elaborated and estimated the average cost per ton of wood chips produced (EUR/t). The values are shown in the table below (Table 4).

However, having considered hypothetical cultivation systems, the present review is limited to resume the productivity reported in the literature [13]. Considering the application of traditional mechanization (chainsaw, winch, chipper pulled by tractor, and loader), the average gross time range for the felling/staging phase varies from 145.9 h/ha to 72.9 h/ha, for extraction from 19 h/ha to 12.7 h/ha, and for loading 12.9–9.6 h/ha for a total amount of hours ranging from 177.8 h/ha to 95.2 h/ha. Instead, in the case of an advanced mechanization level (harvester, forwarder, chipper on forwarder and loader), the average gross utilization time is considerably reduced: during the felling/staging phase, the hours per hectare per worker vary from 14.5 to 6.6 h/ha, for the extraction from 7.8 to 6.2 h/ha, and finally for the load from 6.7 to 4.9 h/ha for a total range from 29 to 17.7 h/ha.

**Table 2.** Technical and economic elements adopted for the calculation of the hourly cost of machines according to the phases of use and the level of mechanization adopted.

| | | Felling/Staging | | | Skidding | | Shredding | | Load | |
|---|---|---|---|---|---|---|---|---|---|---|
| **Level of Mechanization** | | **Traditional (T)** | **Advanced (A)** | **Pushed (P)** | **T/A** | **P** | **T/A** | **A/P** | **T/A/P** | |
| **Technical and Economic Elements** | **Symbols** | **Chainsaw** | **Feller Buncher** | **Harvester** | **Tractor with Winch** | **Forwarder** | **Chipper Pulled by Tractor** | **Chipper on Forwarder** | **Charger** | **Equations or Values** |
| New value (EUR) | $V_0$ | 946 | 100.000 | 380.000 | 50.000 | 220.000 | 100.000 | 520.000 | 80.000 | |
| Recovery value (EUR) | $V_n$ | 601.71 | 22.130 | 84.094 | 11.065 | 48.686 | 22.130 | 115.076 | 17.704 | $V_0*0.86^Y$ |
| Useful life (years) | Y | 3 | 10 | 10 | 10 | 10 | 10 | 10 | 10 | |
| Annual use (h) | H | 700 | 1000 | 1000 | 1000 | 1000 | 800 | 800 | 800 | |
| Nominal power (kW) | P | 3.5 | 75 | 173 | 113 | 140 | 360 | 404 | 88 | |
| Interest rate (%) | i | 4 | 4 | 4 | 4 | 4 | 4 | 4 | 4 | |
| Average fuel cost (EUR/L) | Cfu | 1.89 | 1.45 | 1.45 | 1.45 | 1.45 | 1.45 | 1.45 | 1.45 | |
| Fuel density (kg/L) | Fd | 0.7 | 0.86 | 0.86 | 0.86 | 0.86 | 0.86 | 0.86 | 0.86 | |
| Average lubricant cost (EUR/L) | Clu | 4.5 | 9 | 9 | 9 | 9 | 9 | 9 | 9 | |
| Hourly fuel consumption (L/h) | Chf | 0.68 | 9.59 | 22.12 | 14.45 | 17.9 | 46.05 | 51.67 | 11.25 | |
| Hourly lubricant consumption (L/h) | Chl | 0.32 | 0.32 | 0.73 | 0.26 | 0.55 | 1.4 | 1.7 | 0.26 | |
| Storage space (m²) | Ss | 0.6 | 25.6 | 48.4 | 28.2 | 32.9 | 30 | 41.9 | 18.4 | le*wi*1.5 (*) |
| Shelter construction cost (EUR/m²) | Ccs | 500 | 500 | 500 | 500 | 500 | 500 | 500 | 500 | |
| Coeff. miscellaneous expenses | C.me | 0.001 | 0.005 | 0.01 | 0.003 | 0.005 | 0.005 | 0.01 | 0.001 | 0.001–0.01 |
| Coeff. insurance | C.i | 0.02 | 0.02 | 0.02 | 0.02 | 0.02 | 0.02 | 0.02 | 0.02 | 0.01–0.03 |
| Coeff. overheads | C.o | 0.001 | 0.005 | 0.005 | 0.005 | 0.005 | 0.005 | 0.005 | 0.001 | 0.001–0.01 |
| Coeff. storage costs | C.cs | 0.02 | 0.05 | 0.05 | 0.05 | 0.05 | 0.05 | 0.05 | 0.05 | 0.02–0.05 |

(*) le and wi indicate the length and width of the machine, respectively [62].

**Table 3.** Hourly operating costs of machines (including labor) at different stages of use of the poplar weight, considering different levels of mechanization.

| | | Felling/Staging | | | Skidding | | Shredding | | Load | |
|---|---|---|---|---|---|---|---|---|---|---|
| **Level of Mechanization** | | **Traditional (T)** | **Advanced (A)** | **Pushed (P)** | **T/A** | **P** | **T/A** | **A/P** | **T/A/P** | |
| **Cost Items** | **Symbols** | **Chainsaw** | **Feller Buncher** | **Harvester** | **Tractor with Winch** | **Forwarder** | **Chipper Pulled by Tractor** | **Chipper on Forwarder** | **Charger** | **Equations Used** |
| | | | | | VARIABLE COSTS (VCh) | | | | | |
| Maintenance | MA | 0.5 | 6.0 | 22.8 | 3.0 | 13.2 | 6.0 | 31.2 | 4.8 | $V_0 \times \mu$ (*) |
| Fuel cost | FC | 1.29 | 13.91 | 32.07 | 20.95 | 25.96 | 66.77 | 74.92 | 16.31 | $Cfu \times Chf$ |
| Lubricant cost | LC | 0.26 | 2.78 | 6.41 | 4.19 | 5.19 | 12.6 | 14.98 | 3.26 | $Clu \times Chl$ |

**Table 3.** *Cont.*

| Cost Items | Symbols | Felling/Staging | | | Skidding | | Shredding | | Load | Equations Used |
|---|---|---|---|---|---|---|---|---|---|---|
| Level of Mechanization | | Traditional (T) | Advanced (A) | Pushed (P) | T/A | P | T/A | A/P | T/A/P | |
| | | Chainsaw | Feller Buncher | Harvester | Tractor with Winch | Forwarder | Chipper Pulled by Tractor | Chipper on Forwarder | Charger | |
| Miscellaneous expenses | MIE | 0.01 | 0.1 | 0.38 | 0.05 | 0.22 | 0.06 | 0.65 | 0.1 | $V_0/(Y \times H) \times C.me$ |
| Labor | LA | 15 | 15 | 15 | 15 | 15 | 15 | 15 | 15 | |
| Total VCh | | 17.1 | 37.8 | 76.7 | 43.2 | 59.6 | 100.43 | 136.8 | 39.5 | MA + FC + LC + MIE + LA |
| FIXED COSTS (FCh) | | | | | | | | | | |
| Amortization | AM | 0.16 | 7.79 | 29.59 | 3.89 | 17.13 | 24.33 | 50.62 | 7.79 | $(V_0 - V_n)/(A \times H)$ |
| Insurance | INS | 0.03 | 2.0 | 7.6 | 1.0 | 4.4 | 2.5 | 13.0 | 2.0 | $(V_0 \times C.i)/H$ |
| Storage | STO | 0.01 | 0.64 | 1.21 | 0.71 | 0.82 | 0.09 | 1.31 | 0.57 | $(Ccs \times Ss \times C.me)/H$ |
| Interest | IN | 0.04 | 2.44 | 9.28 | 1.22 | 5.37 | 3.05 | 15.88 | 2.44 | $[((V_0 + V_n)/(2 \times H)) \times i]/100$ |
| Overhead | OH | 0.001 | 0.06 | 0.24 | 0.03 | 0.14 | 0.15 | 0.40 | 0.01 | $(AM + INS + STO + IN) \times C.o$ |
| Total FCh | | 0.60 | 12.93 | 47.92 | 6.85 | 27.86 | 30.12 | 81.21 | 12.81 | AM + INS + STO + IN + OH |
| **TOTAL COSTS (EUR/h)** | | **17.62** | **50.72** | **124.58** | **50.05** | **87.43** | **130.55 \*** | **217.96** | **52.29** | |

(\*) Maintenance costs are estimated as a percentage of the new value. The maintenance coefficients (µ) are between 0.00005 and 0.00006. \* To the cost of chipper pulled by tractor must be added the operating cost of the tractor, which has been estimated at 50 (EUR/h).

**Table 4.** Average cost per ton of wood chips (EUR/t) at different stages of use of the poplar weight, considering different levels of mechanization.

| | Felling/Staging | | Skidding | | Shredding | | Load |
|---|---|---|---|---|---|---|---|
| Level of Mechanization | Traditional (T) | Pushed (P) | T/A | P | T/A | A/P | T/A/P |
| | Chainsaw | Harvester | Tractor with Winch | Forwarder | Chipper Pulled by Tractor | Chipper on Forwarder | Charger |
| TOTAL COSTS (EUR/h) | 17.62 | 124.58 | 50.05 | 87.43 | 130.55 \* | 217.96 | 52.29 |
| PRODUCTIVITY (t/h) | 1.2–2.5 | 12.4–27.1 | 9.8–14.2 | 23–29 | 10.4–14 | 14–18 | 10.4–20.1 |
| **TOTAL COSTS \* (EUR/t)** | **9.52** | **6.30** | **4.17** | **3.36** | **14.80** | **13.62** | **5.03–2.60** |

\* Average of productivity values given in reference [13].

The cost analysis for the final product, in relation to the degree of mechanization, reveals that the implementation of a higher level of mechanization entails a significant reduction in the overall costs of the yard.

Based on the analyses carried out, the adoption of a high level of mechanization leads to a reduction in operating costs per ton from 33.52 EUR/t to 25.88 EUR/t; similarly, the costs per hectare increase from an amount of 4518.8 EUR/ha to 2850.8 EUR/ha.

In addition, it should be considered that the estimates are based on an annual use of the machines at a maximum power of around 800–1000 h; less use would entail considerable cost increases compared to those reported.

According to [4], a team that adopts a push mechanization can use up to 100 ha per year against about 12–15 ha of those who work with traditional yard methods. Therefore, a level of mechanization push is feasible and advantageous only if the technical, logistic, and commercial organization of the enterprise allows us to optimize the potentiality of the means available that must work with continuity over the whole day and the year.

A higher level of mechanization also shows great advantages in terms of productivity. The high degree of mechanization allows for a high level of work organization and an increase in the productivity of the site.

The use of the harvester gives the yard a daily productivity of up to 35 tons per worker compared to 12 tons in the traditional system [4,68]. When compared with a more traditional working methodology, there is a reduction in working time of 117 h per worker, which in terms of productivity, corresponds to 6.86 t/ha per worker as opposed to 1.26 t/ha [69].

These economic and work organization advantages translate into greater company efficiency, including at the budgetary level, as well as an improved psychophysical well-being of workers—well-being which, in turn, results in an increased working capacity and improved productivity [70,71].

Economic and production efficiency, as analyzed in paragraph 4, can be further increased using new innovative machines and new technologies [72,73]. Precision forestry, autonomous and assisted driving, as well as the latest generation of remote sensing, offer significant benefits for forest production and operator safety [71].

It is fair to point out that, for a complete and comparative analysis of the two systems, it would be appropriate to quantify the revenues from the final products (quality wood for industry, in the case of the traditional and wood chip systems in the case of SRC) [74,75].

## 7. Conclusions

From the work carried out, it has been understood how mechanization in the forestry sector and, in particular, in that of poplar cultivation can help to increase economic efficiency to improve productivity and reduce worker accidents, as well as increase the psychophysical well-being of workers.

From the analysis of the data carried out, it can be noted that, given the high initial investment, the implementation of a high level of mechanization entails a significant reduction in the overall costs of the yard, as well as a higher plant productivity and therefore higher profits.

In fact, based on the calculation of the operating costs of the machines implemented by the authors, it seems that the use of a strong mechanization involves a percentage reduction of the average cost per ton of wood chips produced (EUR/t) of 23%, from 33.52 EUR/t to 25.88 EUR/t and a percentage reduction of the average gross cost per hectare of wood chips produced (EUR/ha) of 37%, from an amount of 4518.8 EUR/ha to 2850.8 EUR/ha.

Economic and production efficiency, analyzed in Section 4, can be further enhanced by using innovative new machines and new technologies such as autonomous and assisted driving and latest-generation remote sensing, which offer significant benefits for forest production and operator safety. By using these innovative machines, it is possible to have higher-quality final products, thus guaranteeing broad protection for the consumer who

will be less averse to purchasing and will contribute to increasing the turnover of the market for new renewable energies deriving from woody biomass.

Of course, the European and national institutions must play a key role in encouraging the generational replacement of obsolete agricultural and forestry machinery and in encouraging the use of the latest technologies.

**Author Contributions:** Conceptualization, V.D.S., G.D.D. and A.C.; Data analysis, G.D.D. and V.D.S.; Methodology, V.D.S., G.D.D. and A.C.; Writing original draft, V.D.S. and G.D.D.; Review and editing, E.P., M.M., L.B. and A.C. All authors have read and agreed to the published version of the manuscript.

**Funding:** This research received no external funding.

**Data Availability Statement:** Not applicable.

**Acknowledgments:** This research was carried out within the framework of the Ministry for Education, University, and Research (MIUR) initiative, the "Department of Excellence" (Law 232/2016) DAFNE Project 2023-27, "Digital, Intelligent, Green and Sustainable (acronym: D.I. Ver.So)" and under the General Agreement between Council for Agricultural Research and Economics and University of Tuscia.

**Conflicts of Interest:** The authors declare no conflicts of interest.

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
