# Peer review of "Comparison between Different Mechanization Systems: Economic Sustainability of Harvesting Poplar Plantations in Italy"

_forests, doi:10.3390/f15030397_

Round 1

Reviewer 1 Report

Comments and Suggestions for Authors

The Article deals with the analysis of changes in poplar cultivation (forestry and plantation activities) with the simultaneous costs resulting from the application of modern breeding techniques.

 Comments 

Abstract

This part of the work contains an overly elaborate description of the preliminary demand analysis, lacked a detailed description of the methodology and presentation of the results of the conducted research. Unfortunately, the abstract does not contain the final conclusions of the conducted research, with which it does not encourage the reader to read the article. Necessarily, the authors should rewrite the Abstract.

Introduction

 The presentation of poplar crops should include an indication of the directions for the use of poplar wood in a broader context - that is, furniture, packaging, boards and energy . The economic factor - prices - is not supported by the source of obtaining this data.

The cost of biomass production is already related to the literature, also in terms of economics.

Mechanization of the afforestation and cultivation process is a very important factor in the costs but also in the economic effects of forestry operations.

The article, describing the machinery and equipment used, is based on a wide range of databases and the method of selection of keywords for the search of theses was not indicated.

The article, in terms of the research conducted, is intended to be a review paper, which should involve an extensive literature review.

The significant role of poplar crops as the main source of wood raw material in Italy (45%) is indicated. Poplar plantations are a critical source of biomass in this system.

Mechanization of cultivation and harvesting is based on the principles of sustainable development. What issues of this area have the authors presented in the work?

The process of planting and preparation for planting is presented in great detail.

The topics of biomass harvesting and the cost of processing into post-industrial lumber and woodchips are presented based on the literature.  How is the reference to costs broken down into the labor portion and the cost of using machinery and equipment in the literature?

The form of chipping and harvesting of energy biomass is one of the key issues. What translates into energy value and justification for accepting such poplar biomass for processing?

What poplar varieties find justification in current plantation crop directions? Also with respect to cultivation and harvesting costs? Have the authors conducted such an analysis?

Table 2 refers to the cost of selected equipment. The authors did not indicate how and when they measured and evaluated the costs of machinery work. Only in the text do they report on the cost analysis (literature 57 to 63)

Reference was made to the assumptions of the Green Deal. Traditional mechanization; and advanced mechanization was presented on several models with detailed cost presentation. There was no indication of the source of price data

The description and discussion of the results are not objectionable

Conclusions

  The conclusions correspond with the presented results. However, the part of the description referring to the literature should be in the discussion of the results and not in the conclusions.

Conclusions are not supported by numerical values of the indicator of the impact of mechanization on the reduction of fief production costs in poplar cultivation.

The paper is interesting and of cognitive importance. A broader review of the literature was missing.

Unfortunately, there is a lack of systematization of the results and their optimal use in the discussion.

Reviewer 2 Report

Comments and Suggestions for Authors

Dear Editor and Author/s.

The suggestion seen below is also important.

It is suggested that the calculation method and equations used to determine the values in Table 2 and Table 3 should be detailed. The revisions can be seen in the attached file.

Sincerely Yours.

Reviewer 3 Report

Comments and Suggestions for Authors

The main content of this paper is to emphasize the crucial role of mechanization in promoting the growth of poplar, focusing on various levels of mechanization for poplar harvesting and addressing the associated costs, and there were still several problems as follows:

1. I think figure 1 can be improved appropriately, firstly, the data in the figure is not clear enough, the specific values are not labeled in the figure, also, does the blank part mean that there is no poplar cultivation in these areas, so what is the significance of their existence in the figure?

2. What the review wants to discuss is how mechanization help to increase efficiency and improve productivity in poplar cultivation, but the process of calculating the concluding data given therein is not clear enough. What are the factors that influence the final outcome? What is the formula between these factors?

3. In section 2, whether it makes sense to have a separate section on poplar cultivation? Are changes in the area planted with poplar in Italy related to the development of different mechanical systems? Whether regional estimated area (ha) for poplar cultivation should be discussed along with changes in mechanical systems?

4. Since this is a discussion of the effects of different mechanical systems on poplar cultivation, I think pictures of these mechanical systems should be added to enable the reader to have a more visualized impression.

5. It should be made clearer which data in Tables 1-3 are from the references and which data have been reanalyzed by the authors themselves. These numbers are the keys to this review.

6. In line 47, should "Just think" be expressed differently? 

Comments on the Quality of English Language

Minor editing of English language required

Round 2

Reviewer 3 Report

Comments and Suggestions for Authors

No more comments.

Comments on the Quality of English Language

No more comments.

Author Response

Dear Reviewer,

thank you!